

**The sensitivity of benzene cluster cation chemical ionization mass spectrometry to select**
**biogenic terpenes**
Avi Lavi[1,2], Michael P. Vermeuel[1], Gordon A. Novak[1], Timothy H. Bertram[1, *]
[1]Department of Chemistry, University of Wisconsin, Madison, WI 53706, USA;
[2]Now at: Department of Chemistry, University of California-Riverside, Riverside, CA 92521,
USA;
*Correspondence to: T.H. Bertram, timothy.bertram@wisc.edu
**Abstract**
Benzene cluster cations are a sensitive and selective reagent ion for chemical ionization of select
biogenic volatile organic compounds. We have previously reported the sensitivity of a field
deployable chemical ionization time-of-flight mass spectrometer (CI-ToFMS), using benzene
cluster cation ion chemistry, for detection of dimethyl sulfide, isoprene and alpha pinene. Here,
we present laboratory measurements of the sensitivity of the same instrument to a series of
terpenes, including isoprene, α-pinene, β-pinene, D-limonene, ocimene, β-myrcene, farnesene, α-
humulene, β-caryophyllene and isolongifolene at atmospherically relevant mixing ratios (< 100
pptv). In addition, we determine the dependence of CI-ToFMS sensitivity on the reagent ion
neutral delivery concentration, the instrument electric field strength and water vapor concentration.
We show that isoprene is primarily detected as an adduct ($C_5H_8 \cdot C_6H_6^+$) with a sensitivity ranging
between 4-10 ncps ppt$^{-1}$, that depends strongly on the reagent ion precursor concentration, de-
clustering voltages, and specific humidity (SH). Monoterpenes are detected primarily as the
molecular ion ($C_{10}H_{16}^+$) with an average sensitivity, across the five measured compounds, of $14 \pm$
3 ncps ppt$^{-1}$ for SH between 7 and 14 g kg$^{-1}$, typical of the boreal forest during summer.
Sesquiterpenes are detected primarily as the molecular ion ($C_{15}H_{24}^+$) with an average sensitivity,
across the four measured compounds, of $9.6 \pm 2.3$ ncps ppt$^{-1}$ that is also independent of specific
humidity. Comparable sensitivities across broad classes of terpenes (e.g., monoterpenes and
sesquiterpenes), coupled to the limited dependence on specific humidity, suggests that benzene
cluster cation CI-ToFMS is suitable for field studies of biosphere-atmosphere interactions.



## 1. Introduction

The annual global emission of biogenic volatile organic compounds (BVOCs) is estimated at 1000 TgC yr$^{-1}$ and exceeds the total VOC emissions from anthropogenic activities (Guenther et al., 2012). Foliage emissions account for 90% of global BVOC emissions, of which isoprene ($C_5H_8$), monoterpenes (MTs; $C_{10}H_{16}$) and sesquiterpenes (SQTs; $C_{15}H_{24}$) are the primary constituents (Guenther et al., 1995). The emission rate and the chemical composition of emitted BVOCs is a complex function of the vegetation species and the wide array of stress factors that it is exposed to (Hallquist et al., 2009; Lang-Yona et al., 2010; Zhao et al., 2017). Atmospheric oxidation of BVOCs results in the formation of low volatility compounds that can lead to new particle formation (Jokinen et al., 2015; Kirkby et al., 2016) and particle growth through secondary organic aerosol formation (Allan et al., 2006; Wiedensohler et al., 2009). Both of these processes impact Earth's radiative budget by scattering solar radiation and/or altering cloud formation and precipitation (Chung et al., 2012). The contribution of different types of BVOCs (e.g., isoprene, MTs and SQTs) to secondary organic aerosols (SOA) differ significantly (Zhao et al., 2017). Therefore, uncertainties in BVOCs emissions present significant issues in estimating net climate forcing (Kerminen et al., 2005; Kulmala et al., 2004). Identification of the chemical composition of the emitted BVOCs and quantification of the surface exchange rates of these compounds are essential for understanding complex and non-linear biosphere-atmosphere interactions.

Chemical ionization mass spectrometry (CIMS) is a commonly utilized selective and sensitive method for *in situ* detection of trace gases (Huey, 2007). The sensitivity and selectivity towards a specific compound or class of compounds having similar functional groups rely on the selection of an appropriate ion (i.e. reagent ion) that reacts with and ionizes the analyte *via* an ion-molecule reaction. For example, iodide ions have been used to measure reactive nitrogen compounds, halogen containing species and oxygenated VOCs (Lopez-Hilfiker et al., 2015; Riedel et al., 2012; Thornton et al., 2010), $CF_3O^-$ has been used for the detection of peroxides and organic nitrates (Crounse et al., 2006), $NO^+$ has been used for the selective detection of primary alcohols and alkenes (Hunt and Harvey, 1975; Hunt et al., 1982), $H_3O^+$ for VOCs and their oxygenated products (Lindinger et al., 1998) and benzene cluster cations for dimethyl sulfide (DMS), isoprene, and terpenes (Kim et al., 2016; Leibrock and Huey, 2000).



The benzene cation clusters spontaneously with neutral benzene *via* attractive, non-covalent
interactions (Chipot et al., 1996; Grover et al., 1987). Leibrock and Huey (2000) and recently Kim
et al. (2016) demonstrated that select VOCs including isoprene, MTs, SQTs and aromatic
compounds can be ionized by benzene cation clusters. Kim et al. studied the parameters that
control the benzene cation cluster distribution $(C_6H_6)^+ \cdot (C_6H_6)_n$ at the operational conditions of the
CI-ToFMS, concluding that, for the specific operating conditions used, the reagent ion within the
ion-molecule reaction chamber was primarily in the form of the benzene dimer or larger clusters
(Kim et al., 2016). This conclusion is in agreement with studies showing that the dissociation
energy of the benzene cation dimer is significantly higher than that of the trimer or larger benzene
cation clusters (Krause et al., 1991), suggesting that ionization in the CI-ToFMS by benzene cluster
cations proceeds primarily through clusters that are at least the size of the benzene cation dimer.
The ionization mechanism for a given analyte (M) with the benzene cation dimer, depends on the
ionization energy (IE) of the analyte. Charge transfer (R1) is expected to be the dominant reaction
for analytes having ionization energies smaller than the benzene dimer (8.69 eV) (Grover et al.,
1987). In cases when the analyte IE is higher than that of benzene cation dimer, charge transfer is
thermodynamically unfavored and adduct formation (R2) or ligand exchange (R3) are the sole
modes of ionization. The ligand exchange product (R3) was previously reported for isoprene,
dimethyl sulfide and select alkenes, however the reaction pathway is not known (Kim et al., 2016;
Leibrock and Huey, 2000).

80         $(C_6H_6)_2^+ + M \rightarrow M^+ + 2C_6H_6$          (R1)

81         $(C_6H_6)_2^+ + M \rightarrow M^+ \cdot (C_6H_6)_2$           (R2)

82         $(C_6H_6)_2^+ + M \rightarrow M^+ \cdot (C_6H_6) + C_6H_6$       (R3)

The low IE of benzene clusters (8.69 eV for the dimer and even smaller for larger benzene cation
clusters) (Grover et al., 1987; Shinohara and Nishi, 1989) is a major advantage in the quantification
of monoterpenes or larger volatile organic compounds such as sesquiterpenes. The IE of these
compounds is slightly smaller than that of the benzene dimer (e.g. 8.3 eV for β-caryophyllene
(Novak et al., 2001)) and the minimal excess energy in charge transfer reactions results in limited
fragmentation. For example, approximately 60% of β- caryophyllene was detected in its molecular



ionic form (M$^+$) in comparison to significant fragmentation observed by proton transfer reaction
mass spectrometry (PTR-MS) (Kim et al., 2014; Kim et al., 2009).
The field deployable CIMS that utilizes a time-of-flight mass analyzer (ToFMS), previously
described by Kim et al. combines the efficient production and transmission of ions at high pressure
(e.g. 75 mbar) with the high ion duty cycle of orthogonal extraction ToFMS (Bertram et al., 2011).
This instrument configuration is highly sensitive and capable of measuring and logging mass
spectra (10-800 $m/Q$) at rates higher than 10 Hz (Bertram et al., 2011). These benefits make CI-
ToFMS highly applicable for studying atmospheric exchange processes of trace gases at the air-
ocean interface that require fast response rates (Kim et al., 2014). However, at these pressures, the
distribution of benzene clusters and their associate ion-molecule reactions times are not well
constrained. Unlike PTR-MS, it is not possible to directly derive the analyte mixing ratio from
laboratory studies of the ion-molecule kinetics (reaction rates) that are conducted at lower pressure
in which both the reaction times and cluster distribution have been previously determined. As such,
quantitative analysis of atmospheric trace gases using high pressure CIMS necessitates either a
direct or empirical calibration for each analyte as a function of the atmospheric conditions (e.g.
humidity or temperature).
In what follows, we build on earlier studies in our group (Kim et al., 2016), which described the
use of benzene cluster cations as a reagent ion for the detection and quantification of dimethyl
sulfide, isoprene, and α-pinene. At the time of Kim et al. (2016), it was not known if: 1)
$C_6H_6 \cdot (C_6H_6)_n^+$ ion chemistry was equally sensitive to all monoterpene compounds, 2) the
dependence of CI-ToFMS sensitivity on specific humidity for a broad range of monoterpenes and
sesquiterpenes, and 3) the source of organic impurities in the reagent ion delivery. Here, we address
each of these topics.
In this paper, we describe a high purity liquid benzene source, which permits operation of the CI-
ToFMS at higher reagent ion concentrations. We discuss the sensitivity of benzene cluster cation
chemistry to a select number of terpenes at atmospherically relevant mixing ratios (<500 pptv).
We report on the effect of atmospheric water vapor and the neutral benzene reagent ion precursor
concentration on CI-ToFMS sensitivity to select terpenes (isoprene, α- and β-pinene, D-limonene,
β-myrcene, ocimene, farnesene, isolongifolene, α-humulene and β-caryophyllene). We
demonstrate the effect of a new set of applied voltages with softer de-clustering power on the




observed cluster distribution in the instrument and discuss the effects of the RF only quadrupole
on ion transmission and its contribution to the de-clustering power of the instrument.
**2. Experimental**
2.1 Materials
The following analytes were purchased from Sigma-Aldrich and used with no further purification:
isoprene, α-pinene, β-pinene, D-limonene (≥99%), β-myrcene (96.2%), ocimene (97.0%, as a
mixture of isomers), farnesene (>90.0%, as a mixture of isomers) α-humulene (>96.5%), β-
caryophyllene (≥98.5%), isolongifolene (≥98.0%, as a mixture of isomers), benzene (≥99.5%) and
chloroform-d (99.8 atom % D). A compressed gas cylinder of 0.184 ppm of DMS-$d_3$ in $N_2$ was
purchased from Praxair, USA. Water was supplied from a Milli-Q system at 18.2 MΩ·cm.
Nitrogen was used from a UHP liquid $N_2$ dewar (Airgas). UHP (99.999%) oxygen cylinders were
purchased from Airgas.
2.2 Chemical Ionization Mass Spectrometer
The detailed description of the CI-ToFMS (Tofwerk AG, Switzerland and Aerodyne Research
Inc., USA) and its performance are discussed in Bertram et. al.(Bertram et al., 2011) In brief,
reagent ions are generated by passing 10 sccm of UHP $N_2$ over the headspace of a liquid benzene
reservoir contained in a stainless steel bottle. Benzene vapor is diluted with 2.2 slpm of $N_2$, prior
to delivery to the $^{210}$Po source. The benzene vapor mixing ratio is estimated from the dilution ratio
and benzene vapor pressure. In the experiments discussed here, we varied the benzene
concentration between 60 and 360 ppm. A combination of stainless steel and Teflon tubing was
used to transfer benzene vapors to minimize extraction of organic compounds from the tubing.
Following dilution, benzene vapor flows through a 10 mCi α emitting radioactive $^{210}$Po source
(NRD 2021–1000). The collision of α-particles with $N_2$ results in the formation of $N_2^+$ ions that
ionize the benzene clusters (Dondes et al., 1966). The analyte sample is mixed with the formed
benzene cluster cations at the ion-molecule reactor (IMR) held at 75mbar. At this pressure, the
estimated analyte residence time in the IMR is 100 ms. The reagent and product ions are
transmitted from the IMR chamber into a collisional dissociation chamber (CDC, P=2 mbar)
equipped with a RF only ion-guide quadrupole, followed by a subsequent chamber (P=1.4 x 10$^{-2}$
mbar) in which a second RF-only quadrupole is used to focus the ion beam. The ion beam is then



guided by a further set of ion optics to the entrance point of the extraction region of the compact
time of flight mass analyzer (Tofwerk AG, Switzerland).
2.3 Liquid Calibration Unit
A custom liquid calibration system was developed to deliver known, atmospherically relevant
mixing ratios (< 500 pptv) of gas-phase terpenes to the CI-ToFMS. The liquid calibration system
uses a syringe pump to continuously evaporate known quantities of solution into a heated carrier
gas flow, generating known mixing ratios of select terpenes. To produce trace concentrations of
each analyte, the standard liquid material was diluted in-series with chloroform-d using a set of
calibrated auto pipettes. Chloroform-d was chosen due to its solvent properties and low boiling
point (61°C) that enhances the evaporation of the analyte. Due to its ionization energy (IE > 11 eV
(Bieri et al., 1981)), higher than that of benzene cation clusters, it was expected that chloroform
would not be ionized and would have negligible impact on the benzene cluster cation ionization
mechanisms. To assess this, mass spectra were recorded for solutions containing solely deuterated
chloroform for a variety of different pump flows from 0 to 5μl min$^{-1}$. We did not observe the
molecular cation of chloroform-d (CDCl$_3^+$, 120 $m/Q$) and only very small signatures of the
fragments (at 48, 84 or 86 $m/Q$) were observed (Figure 1), consistent with the IE of chloroform-d
being higher than that of the reagent ions (11.37 ± 0.02 eV compared with 8.69 eV) (Grover et al.,
1987) (Werner et al., 1974). It was also determined that concentration of deuterated chloroform
did not interfere with reagent ion or water cluster signal intensities.
To evaporate the analyte solution, a controlled amount (0-5μl min$^{-1}$) of the analyte solution was
delivered by a syringe pump (Harvard Apparatus, model 11) *via* PEEK tubing (Upchurch
scientific) into a heated carrier stream resulting in CDCl$_3$ mixing ratios from 60-300 ppmv. A
synthetic 80:20 N$_2$:O$_2$ mixture was used as zero air and heated by an in-line gas heater (Omega,
AHP-3741). The temperature of the zero air flow at the point of intersection with the PEEK tubing
was kept at 80°C via a PID temperature controller (Omega, CN9300). Excess zero air flow was
used to ensure an overflow of the CIMS inlet. The trace concentration of the evaporated analytes
and the elevated temperature in front of the inlet (ca. 50°C) helped to prevent re-condensation of
the analyte on the inlet tubing. Humidified zero air was generated by passing a fraction of the total
flow through the head space of a water reservoir. The relative humidity (RH) of the total air flow



was measured using a relative humidity sensor (Vaisala, HMP110), calibrated using the procedure
described in Greenspan (1977).
The sensitivities reported in this paper are presented in normalized counts per second per pptv
(ncps·pptv$^{-1}$). We normalized the analyte ion count-rates by the sum of the benzene cation
monomer (78 $m/Q$) and dimer (156 $m/Q$) count rates to a reference of $1 \times 10^6$ counts per second of
total reagent ion signal in order to account for changes in ion transmission and generation over
time. Sensitivities are calculated as the slope of the linear fit of each calibration curve of 5-7 steps

184    (

Figure **2**). Error bars are the standard deviation of repeated triplicate measurements. The
performance of the liquid evaporation technique was validated by comparing the sensitivity to
dimethyl-1,1,1-d$_3$ sulfide (Praxair certified compressed gas standard, 0.184 ppm ±10%) diluted by
zero air to a desired mixing ratio, with that of a diluted nebulized solution of DMS. The slope of
the linear fit for calibration measurements from the pressurized cylinder (DMS-d3, 65 $m/Q$) and
the solution (DMS, 62 $m/Q$) agreed to better than 10%.

**3. Results and Discussion**
3.1 Benzene Cluster Cation Mass Spectra
The CI-ToFMS mass spectra, obtained while overflowing the inlet with nominally dry zero air is
shown in Figure 3a. To maximize the transmission of weakly bound ion-molecule adducts, we
operated the instrument in all of the experiments described here with a minimal applied electric
field between the instrument inlet and the entrance of the second RF-only quadrupole ion guide.
The two primary peaks in the mass spectrum correspond to the benzene cation ($C_6H_6^+$; 78 $m/Q$)
and the benzene cation clustered to a single, neutral benzene ($C_6H_6^+ \cdot (C_6H_6)$; 156 $m/Q$), where
$C_6H_6^+$ and $C_6H_6^+ \cdot (C_6H_6)$ combined account over 90% of the total ion current (TIC) for a benzene
neutral concentration of 300 ppm. Benzene cation clusters larger than the dimer were not observed,
as expected from their dissociation enthalpy, which is significantly smaller than that of the benzene
cation clustered with a single neutral benzene molecule (Krause et al., 1991). The observed mass
spectrum indicates significant ion intensity at 39, 50, 51, and 52 $m/Q$ that are attributed to the
dissociation of the molecular ($C_6H_6^+$) ion into its fragments $C_3H_3^+$, $C_4H_2^+$, $C_4H_3^+$, and $C_4H_4^+$,
accounting for ca. 5% of TIC. The fragmentation may result from the interaction of $N_2^+$, α-particles
or electrons with benzene clusters in the ion molecule reaction region (Lifshitz and Reuben, 1969;



Talebpour et al., 2000). For comparison, a similar spectrum is shown in Figure 3b, using the same
benzene neutral concentration and operating voltages, but without the first RF-only quadrupole
ion guide. In this mode of operation, the total ion current is reduced by over 95%, and $C_6H_6^+$ and
$C_6H_6^+\cdot(C_6H_6)$ are nearly equal in intensity, highlighting that benzene cluster collisional
dissociation is occurring within this region. Even with the first RF-only quadrupole off, the n=2
cluster ($C_6H_6^+\cdot(C_6H_6)_2$; 234 $m/Q$) was not observed. Of notable absence (< 1% TIC) in both Figures
3a and 3b are the organic contaminants (92, 106, and 120 $m/Q$) previously attributed to alkyl
substituted benzene and protonated water clusters ($H_3O^+\cdot(H_2O)_n$; 19, 37, 55, and 73 $m/Q$) that were
present at high abundance (>10% of TIC) in Kim et al. (2016). It was postulated in Kim et al., that
the source of the organic contaminants was the benzene compressed gas cylinder, as their
combined contribution to TIC scaled with the neutral benzene concentration. It was also noted that
low benzene neutral concentrations led to elevated water cluster abundance. This resulted in an
optimum benzene neutral concentration of 10 ppm, to balance the contributions from organic
contaminants and water clusters. Here, we eliminate the organic contaminants through the use of
a high purity benzene liquid source permitting operation at higher neutral benzene concentrations
(> 300 ppm). As discussed in section 3.2, this has critical advantages for the detection of analytes
such as isoprene, and effectively eliminates competing ion chemistry stemming from protonated
water clusters.
It what follows we assess the CI-ToFMS sensitivity to a series of terpenes, including isoprene, α-
pinene, β-pinene, D-limonene, ocimene, β-myrcene, farnesene, α-humulene, β-caryophyllene, and
isolongifolene at atmospherically relevant mixing ratios (< 100 pptv) and determine the
dependence of CI-ToFMS sensitivity on the reagent ion neutral delivery concentration (section
3.2) and water vapor concentration (section 3.3).
3.2 Impact of Benzene Neutral Concentration on Terpene Sensitivity
We examined the impact of the benzene reagent ion precursor concentration on terpene sensitivity
in nominally dry zero air for benzene neutral concentrations between 60-300 ppm. For the selection
of monoterpenes and sesquiterpenes studied here, there was no indication that instrument
sensitivity was dependent on the neutral benzene reagent ion precursor concentration between 60–
300 ppm (Figure 4 a-b). In Figure 4a-c, the reported sensitivity for each terpene is normalized to
that measured at a benzene neutral concentration of 300 ppm. Unlike MTs and SQTs, the



sensitivity of the isoprene benzene adduct ($C_6H_6^+ \cdot C_5H_8$; 146 $m/Q$) strongly depends on the benzene
concentration below 200 ppm (Figure 4 c) and therefore all the measurements in this study, were
conducted at 300 ppm benzene. The cause for this dependence in benzene concentration is unclear
as the exact mechanism for $C_6H_6^+ \cdot C_5H_8$ formation is unknown. It should also be noted that the
sensitivity to DMS is independent of benzene concentration. Based on these analyses, we suggest
that future studies utilizing benzene ion chemistry operate at neutral benzene reagent ion precursor
concentrations of 300 ppm, generated from a high purity liquid source.

3.3 Impact of Specific Humidity on Sensitivity
3.3.1 Isoprene
In these experiments, the specific humidity (SH) was varied between 0 and 14 g kg$^{-1}$, equivalent
to 0-80% RH at 23°C, to assess its effect on the sensitivity. Our reported "nominally dry" cases
correspond to 0.7% RH or ca. 0.01 g kg$^{-1}$ SH. As shown in Figure 5, the sensitivity of the CI-
ToFMS to isoprene ($C_6H_6^+ \cdot C_5H_8$; 146 $m/Q$) displays a strong, non-linear dependence on SH.
Instrument sensitivity increases with increasing SH, reaching a maximum value of 10 ncps·ppt$^{-1}$
at 4 g kg$^{-1}$ (25% RH at 23°C), then decreases significantly at higher humidity. Surprisingly, we
observed a linear correlation ($R^2 > 0.95$) between the protonated water tetramer signal (73 $m/Q$)
and the delivered isoprene mixing ratio at constant SH that was not observed for smaller protonated
water clusters (Figure 6). The apparent sensitivity, derived from the slope of the linear-least
squares fit of the observed water tetramer signal *vs.* delivered isoprene concentration, increases
with increasing specific humidity above 2 g kg$^{-1}$ (Figure 5). We reiterate that Figure 5 does not
show the protonated tetramer signal as a function of SH, but the *sensitivity* of the 73 $m/Q$ signal to
the delivered isoprene mixing ratio as shown in Figure 6. The decreased sensitivity to isoprene
adduct and increase in water tetramer signal with isoprene mixing ratio are unlikely the result of
the formation of water protonated clusters *via* charge transfer reaction with benzene cations since
the IE of water is significantly higher than that of the benzene dimer (12.62 and 8.69 eV
respectively). Since the formation of water tetramer clusters increases with isoprene mixing ratio
and humidity, it is suggested that the interaction between water clusters and isoprene-benzene
adducts in the IMR results in a charge exchange from the isoprene-adduct to the water tetramer in
a similar way that was previously described between benzene cation and water clusters. For
example, Ibrahim *et al*. (2005) showed that the IR spectra of benzene-water ion clusters





$[(H_2O)_n \cdot C_6H_6]^+$ where (n $\geq$ 4) resembles that of protonated water clusters and suggested that the
charge is held by the water molecules, such clusters that are likely to be formed in the IMR are
expected to be broken apart in the ion optics. It is likely that the observed trends of the humidity
dependent sensitivity of isoprene and water tetramer signal also results from a similar formation
and de-clustering in our CI-ToFMS.
3.3.2 Monoterpenes
The dependence of monoterpene sensitivity on SH is shown in Figure 7 for the molecular ion
($C_{10}H_{16}^+$; 136 $m/Q$).  Instrument sensitivity under nominally dry conditions displays a wide range
of sensitivities, that are species dependent (4.8 to 21.0 ncps·ppt$^{-1}$). At high specific humidity,
sensitivities converge significantly (9.5 to 15.0 ncps·ppt$^{-1}$). The observed dependence in the α-
pinene sensitivity on SH reported here is counter to that previously reported by our group in Kim
et al. (2016). This is attributed to the different instrument operational configuration used here (e.g.,
high concentration and purity benzene reagent ion precursor and low electric field strengths).
The humidity dependent sensitivity of D-limonene is anomalous compared with the other
monoterpenes studied, where the CI-ToFMS sensitivity to D-limonene decreases by a factor of 4
over the studied humidity range. The gradual and systematic decrease of the sensitivity suggests
that the ionization of D-limonene by charge transfer is not the only ionization mechanism and/or
that the D-limonene cation is subjected to subsequent reactions which results in the formation of
other detectable ions. We calculated the calibration curves of each of the recorded mass-to-charge
ratios to identify product ions that showed: 1) high correlation with the delivered D-limonene
mixing ratio ($R^2 > 0.98$) and 2) the contribution to the total sensitivity (i.e. slope) was higher than
1 ncps ppt $^{-1}$. A representative normalized calibration curve of the three ions (135, 136, and 168
$m/Q$) that met these criteria is presented in Figure 8. The peak at 168 $m/Q$ ($C_{10}H_{16}O_2^+$) is attributed
to either a D-limonene-$O_2$ adduct or a D-limonene oxidation product (e.g. limonene epoxide). The
peak at 135 $m/Q$ ($C_{10}H_{15}^+$) represent the $[M-1]^+$ product, perhaps due to rearrangements of the
molecular ion. The purity of the primary standard was confirmed *via* GC-MS, and comparable
peak ratios were measured when sampling the standard directly, ruling out the potential for the
nebulization process to alter the MS peak ratios. Finally, the $[M+32]^+$ peak intensity is reduced to
baseline by sampling the terpene in nitrogen, suggesting that the $[M+32]^+$ peak is a result of
secondary ion chemistry involving $O_2$. The normalized sensitivity of each of these three peaks



299 decreases with increasing SH (Figure 9), suggesting that water clusters compete or suppress the

300 charge transfer to the contributing ions. The humidity dependent sensitivity of all the studied MTs,

301 calculated as the sum of all their contributing ions, shows lower variability, mostly due to the

302 higher sensitivity to D-limonene when all product ions are accounted for (Figure 10). The

303 variations in the sensitivities between different monoterpenes is small ($14 \pm 3$ ncps ppt$^{-1}$) and

304 instrumental response is largely independent on SH from 4 to 14 g kg$^{-1}$. This range is typical at

305 boreal forests during the summer (Suni et al., 2003).

306 3.3.3 Sesquiterpenes

307 The sensitivities of the CI-ToFMS toward SQTs, detected as the charge transfer product at 204

308 $m/Q$, show minimal dependence on SH between nominally dry conditions and 14 g kg$^{-1}$ (Figure

309 11). Using the same process discussed in section 3.3.2 for identifying other product ions, it was

310 found that 203 and 236 $m/Q$ ($C_{15}H_{23}^{+}$ and $C_{15}H_{24}O_{2}^{+}$) also contributed to product ion intensity.

311 The response of the farnesene and isolongifolene molecular ions and their related contributing ions

312 are presented as examples of SQTs dependence on SH (Figure 12). All three major ions were

313 observed at all measured SHs and in the case of isolongifolene, the normalized response of 203

314 $m/Q$ ($C_{15}H_{23}^{+}$) was higher than the molecular ion (204 $m/Q$, $C_{15}H_{24}^{+}$) over the entire SH range

315 including at nominally dry conditions (Figure 12). At present, we don't have a definitive

316 mechanism for the product ion distribution, but the presence of similar products (i.e. ([M-1]$^{+}$ and

317 ([M+32]$^{+}$) and their humidity dependence suggest that the molecular ions of sesquiterpenes are

318 subjected to similar reactions as MTs which results in a lower signal of the molecular ion. Similar

319 to MTs, the humidity dependent sensitivities of sesquiterpenes calculated as the sum of all

320 contributing ions, lowers the variability in calculated sensitivities (Figure 13). Since the sensitivity

321 is independent of the humidity a general sensitivity to all SQTs of $9.6 \pm 2.3$ ncps pptv$^{-1}$ can be

322 further used for quantification of ambient SQTs.

323

**4. Conclusions**

325 We show that benzene cluster cations are a sensitive reagent ion for chemical ionization of select

326 biogenic volatile organic compounds. We demonstrate that isoprene is primarily detected as an

327 adduct ($C_{5}H_{8}{\cdot}C_{6}H_{6}^{+}$) with a sensitivity ranging between 4-10 ncps ppt$^{-1}$, that depends strongly on





the reagent ion precursor concentration, de-clustering voltages, and specific humidity (SH). This
highlights the importance of continuous infield calibrations for isoprene concentration
measurements. We show that monoterpenes are primarily detected as the molecular ion ($C_{10}H_{16}^+$)
with an average sensitivity, across the five measured compounds, of $14 \pm 3$ ncps ppt$^{-1}$ for SH
between 7 and 14 g kg$^{-1}$, typical of the boreal forest during summer. Sesquiterpenes are detected
primarily as the molecular ion ($C_{15}H_{24}^+$) with an average sensitivity, across the four measured
compounds, of $9.6 \pm 2.3$ ncps ppt$^{-1}$ that is also independent of specific humidity. We suggest that
future studies that utilize benzene cluster cation chemistry use high purity liquid reservoirs and
benzene neutral concentrations at or above 300 ppmv.
**Acknowledgements**
This work was supported by a National Science Foundation (NSF) CAREER Award (Grant No.
AGS-1151430) and the Office of Science (Office of Biological and Environmental Research), U.S.
Department of Energy (Grant No. DE-SC0006431). A.L. gratefully acknowledges support from
the Dreyfus Foundation Environmental Chemistry Postdoctoral Fellowship Program.





**Table 1. Molecular structures for the terpenes characterized in this study.**

isoprene

α-pinene        β-pinene

D-limonene

β-myrcenee

ocimene

isolongifolene        β-caryophyllene        α-humulene

farnesene



**Table 2. Monoterpene sensitivities and dependence on operating and sampling conditions.**

| Compound | Sensitivity[†] (ncps pptv$^{-1}$) (SH = 6.9 g kg$^{1}$) | M$^+$:[M-1]$^+$:[M+32]$^+$ (SH = 0.01 g kg$^{-1}$)[‡] | M$^+$:[M-1]$^+$:[M+32]$^+$ (SH = 6.9 g kg$^{-1}$)[‡] | f(H$_2$O) | f(C$_6$H$_6$) |
|---|---|---|---|---|---|
| α-pinene | 17.9 | 23.9:0.64:0.35 | 17.4:0.21:0.25 | Y | N |
| β-pinene | 18.4 | 14.9:0.28:0.33 | 17.6:0.33:0.39 | N | N |
| D-limonene | 13.6 | 5.4:3.4:8.0 | 3.7:3.0:6.9 | Y | N |
| β-myrcene | 11.5 | 4.6:0.56:0.94 | 8.7:1.1:1.7 | Y | N |
| ocimene | 13.2 | 13.1:1.50:0.29 | 12.4:0.42:0.36 | N | N |

[†]SH = 6.9 g kg$^{-1}$ corresponds to 65 % RH at 15 °C, representative of Boreal regions. The reported
sensitivity includes the contributions from the M$^+$, M-1$^+$, and M+32$^+$ ions.
[‡]Sensitivities (ncps pptv$^{-1}$) at M$^+$, M-1$^+$, and M+32$^+$, is reported for SH = 0.01 and 6.9 g kg$^{-1}$.

**Table 3. Sesquiterpene sensitivities and dependence on operating and sampling conditions.**

| Compound | Sensitivity[†] (ncps pptv$^{-1}$) (SH = 6.9 g kg$^{-1}$) | M$^+$:[M-1]$^+$:[M+32]$^+$ (SH = 0.01 g kg$^{-1}$)[‡] | M$^+$:[M-1]$^+$:[M+32]$^+$ (SH = 6.9 g kg$^{-1}$)[‡] | f(H$_2$O) | f(C$_6$H$_6$) |
|---|---|---|---|---|---|
| farnesene | 10.4 | 7.8:1.3:1.6 | 7.8:1:.1:1.5 | Y | N |
| α-humulene | 8.6 | 5.2:2.6:0.63 | 1:5.3:2.8:0.54 | N | N |
| β-caryophellene | 6.9 | 4.6:1.4:2.2 | 4.0:1.1:1.9 | Y | N |
| isolongifolene | 12.3 | 3.1:7.7:1.2 | 3.4:8.8:0.15 | Y | N |

[†]SH = 6.9 g kg$^{-1}$ corresponds to 65 % RH at 15 °C, representative of Boreal regions. The reported
sensitivity includes the contributions from the M$^+$, M-1$^+$, and M+32$^+$ ions.
[‡]Sensitivities (ncps pptv$^{-1}$) at M$^+$, M-1$^+$, and M+32$^+$, is reported for SH = 0.01 and 6.9 g kg$^{-1}$.

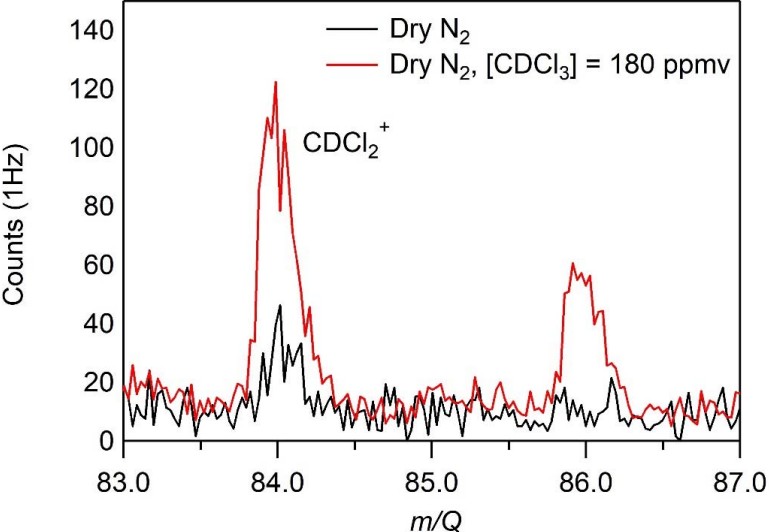

**Figure 1.** CI-ToFMS mass spectrum acquired when overflowing the inlet with excess nitrogen
(black) and for a nebulized solution of chloroform-d at a flow rate of $3\mu l\ min^{-1}$ in a nitrogen carrier
gas (red), where the resulting $[CDCl_3] = 180$ ppmv. No signal was observed above the baseline for
any other fragments or the parent ($CDCl_3^+$, 120 $m/Q$).





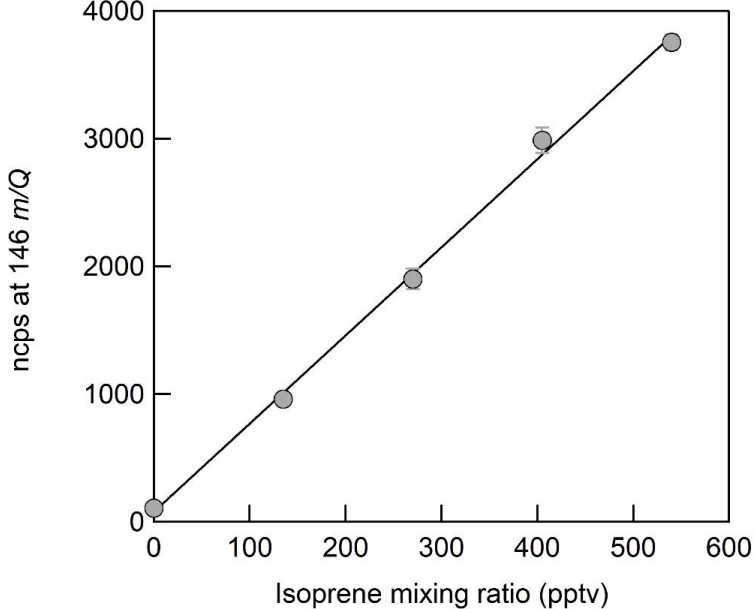

**Figure 2.** CI-ToFMS calibration curve for isoprene, detected as $C_6H_6^+\cdot C_5H_8$ at 146 $m/Q$. The
sensitivity (slope) is 7 ncps, $R^2$=0.99. Error bars represents the standard deviation of the 1Hz
measurements.




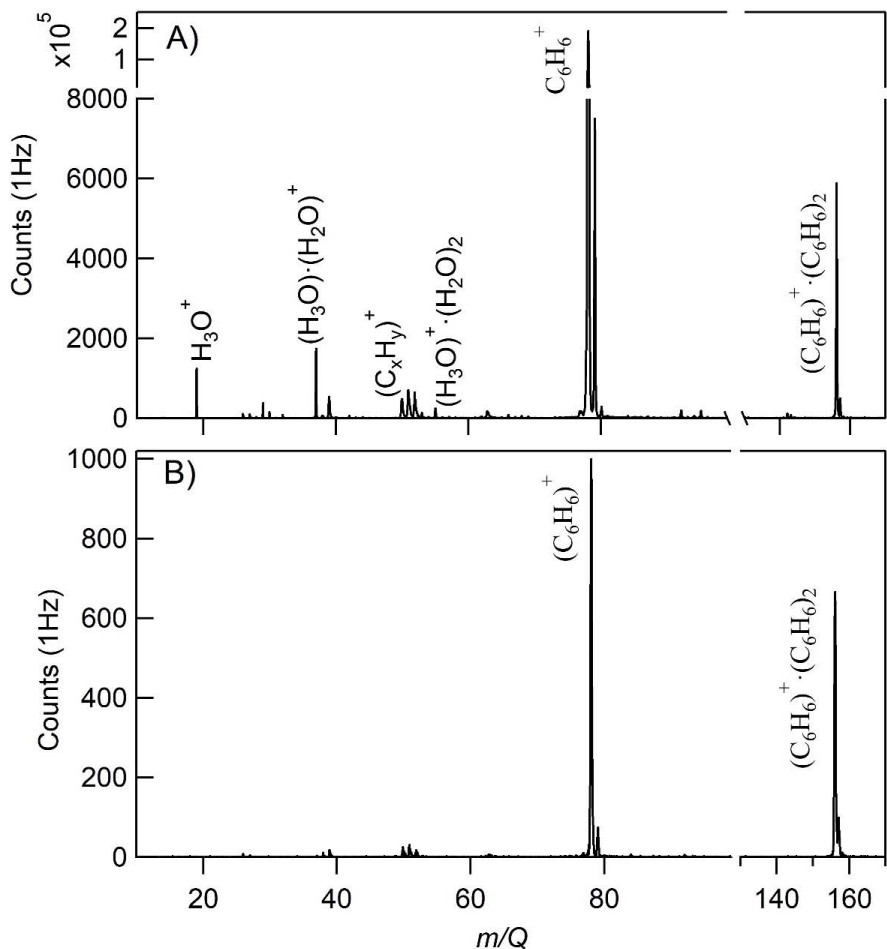

**Figure 3.** a) CI-ToFMS mass spectrum acquired when overflowing the inlet with nominally dry
zero air for a benzene neutral concentration of 300 ppm using a liquid reagent ion delivery and b)
same as in a, but with the first RF-only octupole ion guide turned off, resulting in a much weaker
electric field strength.





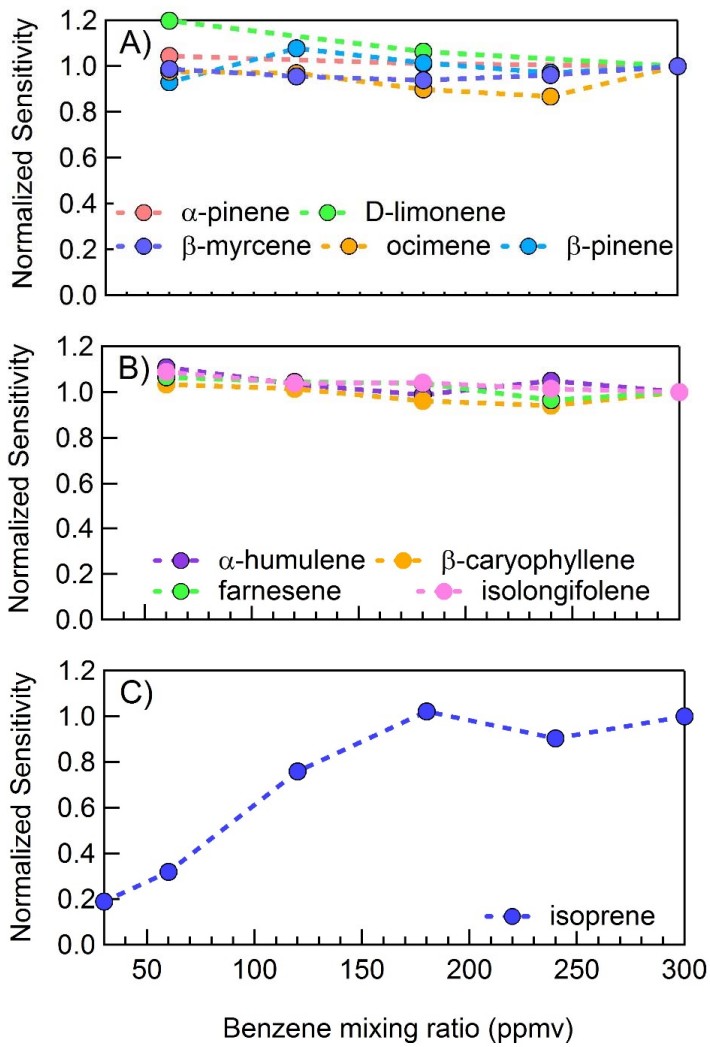

**Figure 4.** CI-ToFMS sensitivity to: a) monoterpenes ($C_{10}H_{15}^+$; 136 $m/Q$), b) sesquiterpenes ($C_{15}H_{24}^+$; 204 $m/Q$), and c) isoprene ($C_6H_6^+ \cdot C_5H_8$; 146 $m/Q$) as a function of benzene neutral concentration normalized to the sensitivity at 300 ppmv neutral benzene. Measurements were conducted in nominally dry zero air.



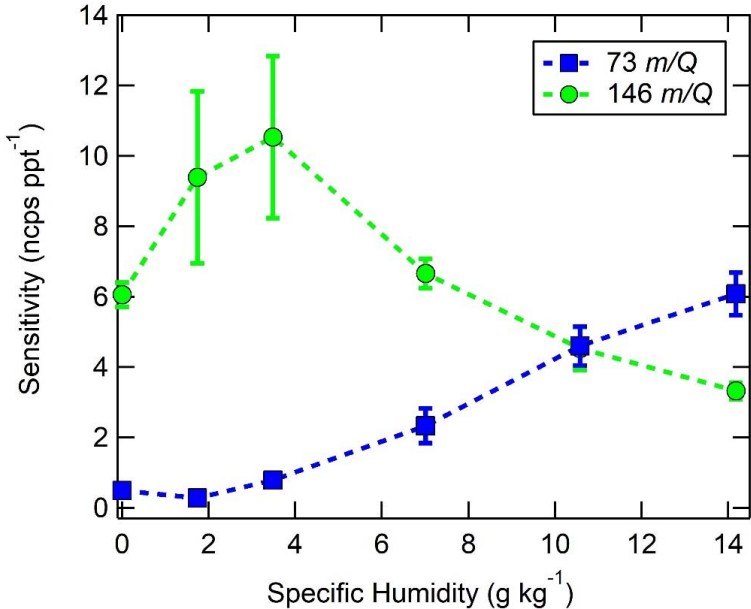

**Figure 5.** Humidity dependent CI-ToFMS sensitivities to isoprene (green circles, $C_6H_6^+\cdot C_5H_8$, 146 $m/Q$), and the protonated water tetramer (blue squares, $H_3O^+\cdot(H_2O)_3$, 73 $m/Q$ ), derived from calibration curves such as those shown in Figure 6. The reported sensitivities are the average of triplicate calibration curves with all linear best fits having $R^2 > 0.98$. Error bars represent the standard deviation of the triplicate calibrations. All calibrations were performed in zero air.



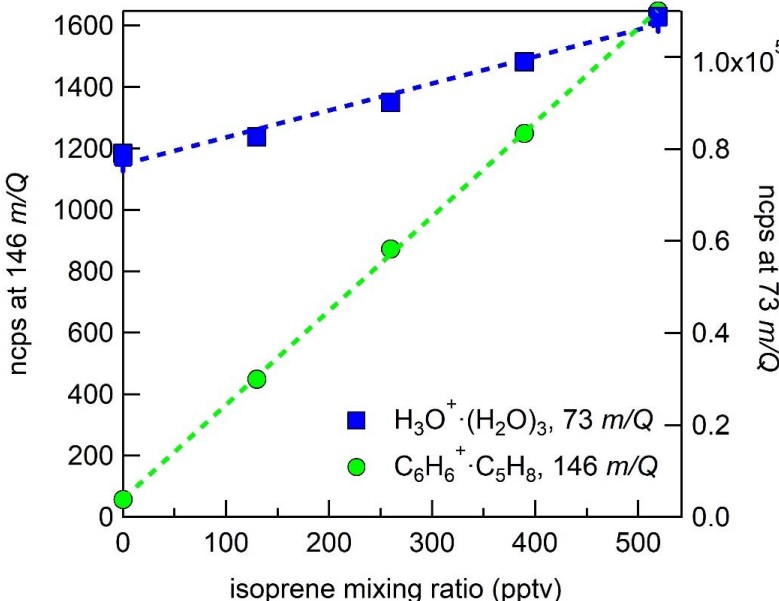

**Figure 6.** CI-ToFMS sensitivity to isoprene, observed as the isoprene-benzene cluster (green
circles, $C_6H_6^+ \cdot C_5H_8$, 146 $m/Q$) and water protonated tetramer (blue squares, $H_3O^+ \cdot (H_2O)_3$, 73 $m/Q$).
Dashed lines are the least square best fit lines ($R^2 > 0.98$). Calibration was performed at SH of 14 g
$kg^{-1}$ in zero air.



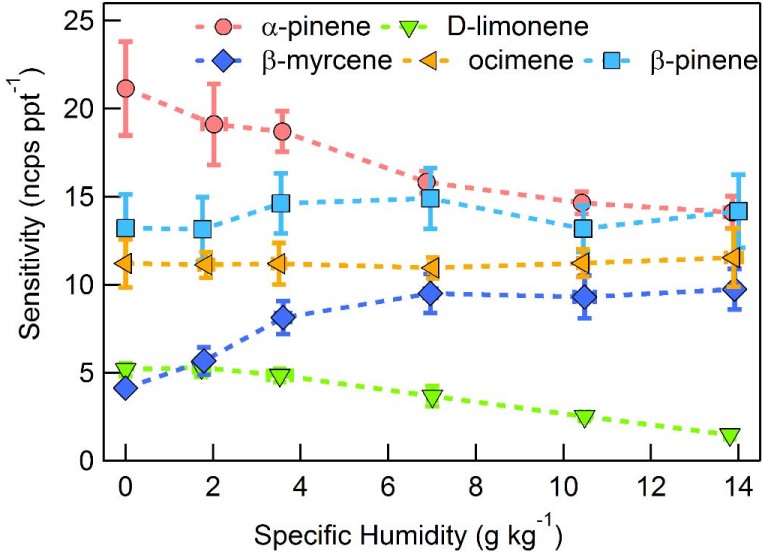


**Figure 7.** Humidity dependent sensitivities to select MTs detected as $M^+$ ($C_{10}H_{16}^+$, 136 *m/Q*). Error
bars indicate the standard deviation of triplicate measurements. All calibrations were conducted in
zero air. Error bars represent the standard deviation of the triplicate calibrations. All calibrations
were performed in zero air.






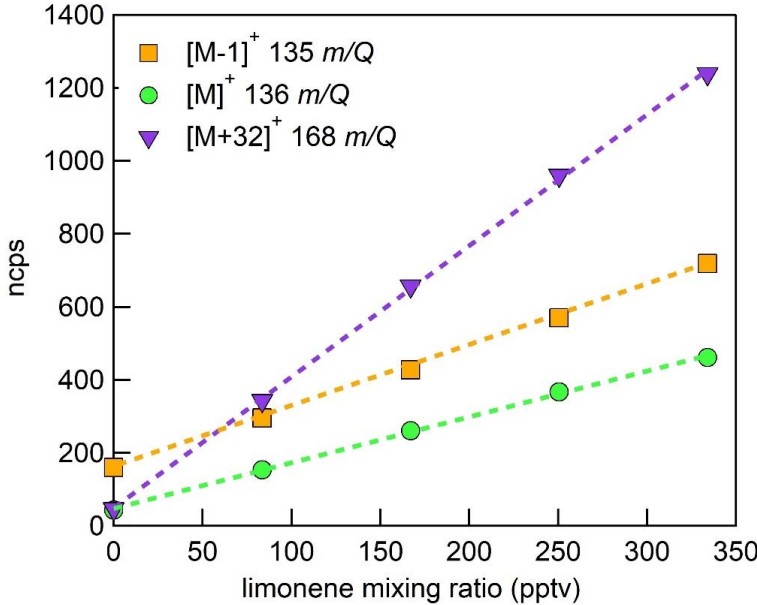


**Figure 8.** Normalized calibration of D-limonene for all major product ions ($C_{10}H_{16}^+$, 136 *m/Q*, green circles), ($C_{10}H_{15}^+$, 135 *m/Q*, orange squares), and ($C_{10}H_{16}O_2^+$, 168 *m/Q*, purple triangles). Calibration was performed in zero air at 14 g kg$^{-1}$ specific humidity (80% RH at 23°C). Dashed lines are least squares best fit lines (all $R^2 > 0.99$).

405




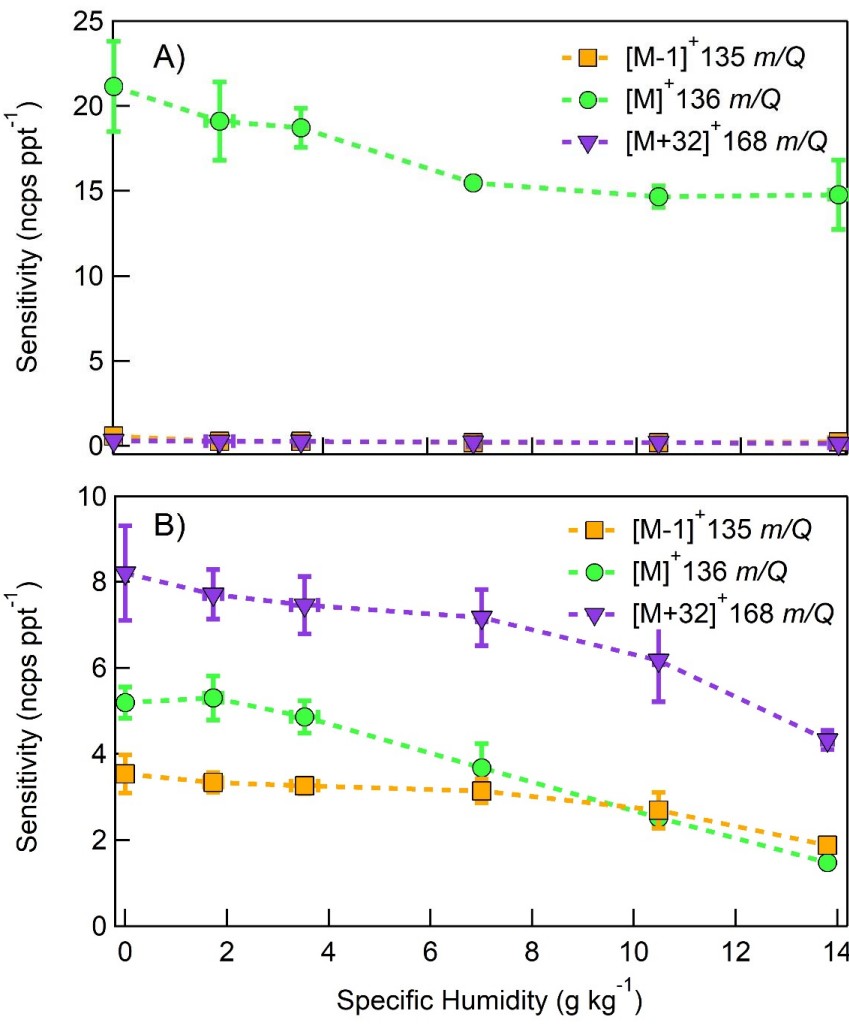

406

**Figure 9.** Humidity dependent, normalized sensitivities to a) α-pinene b) D-limonene for all major
product ions ($C_{10}H_{16}^+$, 136 *m/Q*, green circles), ($C_{10}H_{15}^+$, 135 *m/Q*, orange squares), and
($C_{10}H_{16}O_2^+$, 168 *m/Q*, purple triangles). Error bars represent the standard deviation of the triplicate
calibrations. All calibrations were performed in zero air.

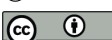




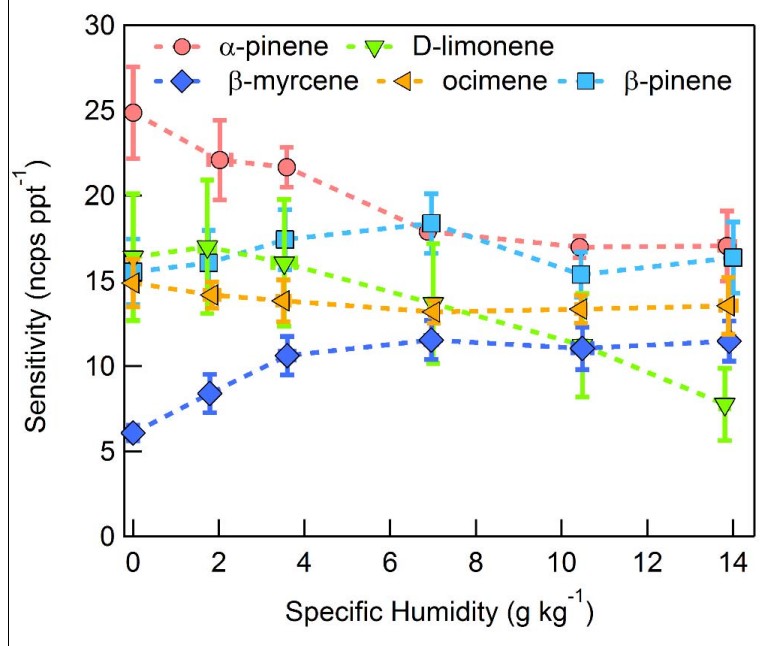


**Figure 10.** Humidity dependent, CI-ToFMS monoterpene sensitivities reported as the sum of all
detected masses (135, 136, and 168 *m/Q*). Error bars represent the standard deviation of the
triplicate calibrations. All calibrations were performed in zero air.





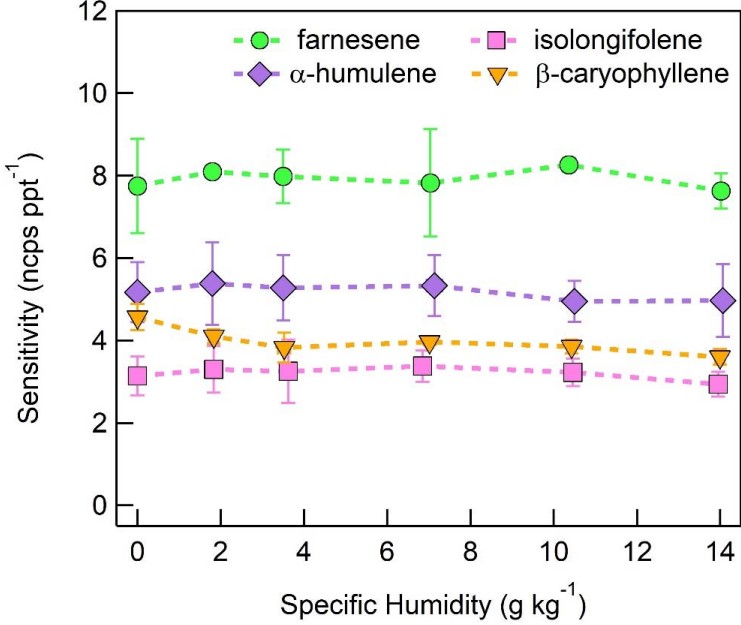


**Figure 11.** Humidity dependent sensitivities of SQTs detected as $C_{15}H_{24}$ (204 *m/Q*). Error bars
represent the standard deviation of triplicate measurements. All calibrations were performed in
zero air.






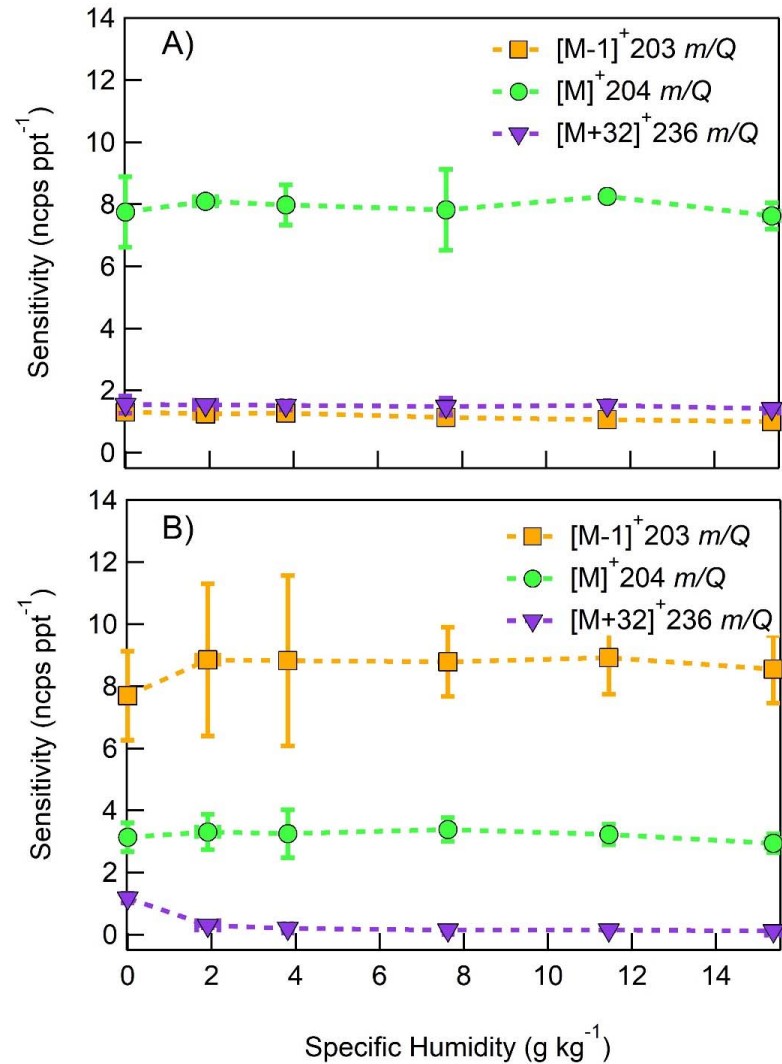

**Figure 12.** Humidity dependent, normalized sensitivities to a) farnesene and b) isolongifolene for
all major product ions ($C_{15}H_{23}^+$, 203 *m/Q*, orange squares), ($C_{15}H_{24}^+$, 204 *m/Q*, green circles), and
($C_{15}H_{24}O_2^+$, 236 *m/Q*, purple triangles). Error bars represent the standard deviation of the triplicate
measurement.

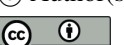



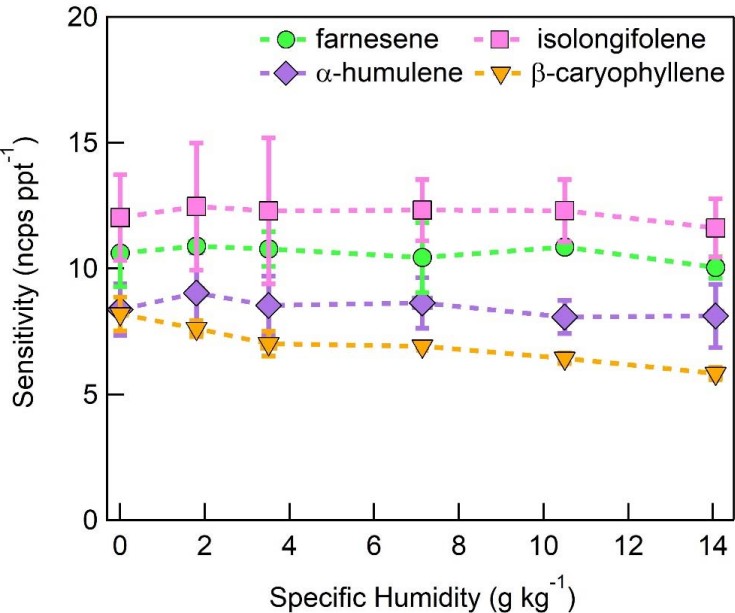


**Figure 13.** Humidity dependent, normalized sensitivities to sesquiterpenes, reported as the sum of the major product ions ($C_{15}H_{23}^+$, 203 *m/Q*), ($C_{15}H_{24}^+$, 204 *m/Q*), and ($C_{15}H_{24}O_2^+$, 236 *m/Q*). Error bars represent the standard deviation of triplicate measurements.




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
