# Peer review of "The sensitivity of benzene cluster cation chemical ionization mass spectrometry to select"

_Atmospheric Measurement Techniques, 2017_

## Referee Comment (RC1) · Anonymous Referee #2 · 13 Feb 2018

Overview

The manuscript by Lavi et al investigates the sensitivity of benzene cluster cation chemical ionization for the detection of a variety of biogenic volatile organic compounds. Importantly, the paper investigates how the sensitivity of this CIMS method depends on the concentration of neutral reagent ion precursor and on water vapor. The work builds on previous work done by the same group using benzene cluster cation CIMS for the detection. Given the growing popularity of CIMS within the atmospheric measurement community, calibration is an area of significant interest. It is important to pursue and publish technical investigations such as this one in order to develop a community

understanding of best practices and limitations of the wide variety of ionization mechanisms being used. I think that this manuscript will be of interest to the community and is appropriate for AMT, however, I feel that there are a few issues that need to be addressed prior to publication.

Major Comment

The manuscript states that a "new set of applied voltages with softer de-clustering power" (line 118) is used here, however there is no information given on what the voltages are or how the voltages compare to those used by the group previously (other than the above vague remark). For instance, there is no discussion about how the relative intensity of the C6H6+ and (C6H6)+(C6H6) compare. Given the current emphasis on voltage scanning within the community (Brophy and Farmer, 2016; Lopez-Hilfiker et al., 2016), I am surprised that such experiments were not carried out here. I think that the inclusion of such experiments would be an asset to the manuscript as it may help provide (at least a basic) explanation of some of the peculiarities that are occurring with isoprene. I am not suggesting that the declustering experiments be performed for all the compounds investigated here; however, the inclusion of a select few would improve the impact of the manuscript. Currently, the only change in the declustering voltages is by turning off the voltages in the first RF-only quadrupole and these results are only shown for sampling zero air (no analyte; Figure 3). While this provides some insight, it is limited given the large drop in ion transmission and the lack of measurement of an analyte. It would be interesting to see how the distribution of [M-1]+, M+, and [M+32]+ changed as a function of voltage scanning for the monoterpenes and sesquiterpenes.

I think it is also an overstatement to say in the abstract (lines 19 & 22) and conclusion (line 328) that the effect of declustering voltages was studied given the limited investigation into this effect.

Minor Comments

Sections 3.1 & 3.2: How does the abundance and ratio between C6H6+ and

(C6H6)+(C6H6) change as the concentration of neutral benzene is increased? Does this provide any insight into the isoprene results?

I do not see where the text references the tables. Including a reference to the table in the text would be beneficial since the tables show that the M+, [M-1]+, and [M+32]+ ions were investigated for all the compounds and not just those discussed in depth in the text.

Technical

Line 14: replace alpha with the correct symbol

Line 57-58: The references for NO+ and H3O+ seem a bit strange as the references refer only to early work and not more recent work, while the references for other ionization mechanisms contains more recent work. For instance, the absence of Koss et al. (2016).

Line 185: "error bars are the standard deviation of repeated triplicate measurements" but caption of figure 2 "error bars represent the standard deviation of the 1 Hz measurements."

Line 209: "Without the first RF-only quadrupole" Is it removed or just the voltages turned off. If the voltages was it all the voltages or just the RF component?

Line 330-331: The wording here makes it appear like a recommendation to only use C10H16+ for quantification of monoterpenes. However, the average sensitivity appears to be that obtained using the M+, [M-1]+, and [M+32]+ ions (Figure 10) and not just the M+ ion (Figure 7). What recommendation do the authors provide for quantification of monoterpenes?

Lines 333-334: Same comment as above expect for sesquiterpenes.

Table 1: I don't find this particularly useful since there is no in depth exploration of how structure might relate to the different observed sensitivities. To be clear, I am

not suggesting that such an explanation be incorporated; I do not think it would be appropriate given the extent of the data.

Figure 3: The peak at m/z 156 should be labeled (C6H6)+ (C6H6)

References

Brophy, P. and Farmer, D. K.: Clustering, methodology, and mechanistic insights into acetate chemical ionization using high-resolution time-of-flight mass spectrometry, Atmos Meas Tech, 9(8), 3969–3986, doi:10.5194/amt-9-3969-2016, 2016.

Koss, A. R., Warneke, C., Yuan, B., Coggon, M. M., Veres, P. R. and de Gouw, J. A.: Evaluation of NO+ reagent ion chemistry for online measurements of atmospheric volatile organic compounds, Atmos Meas Tech, 9(7), 2909–2925, doi:10.5194/amt-9-2909-2016, 2016.

Lopez-Hilfiker, F. D., Iyer, S., Mohr, C., Lee, B. H., D'Ambro, E. L., Kurtén, T. and Thornton, J. A.: Constraining the sensitivity of iodide adduct chemical ionization mass spectrometry to multifunctional organic molecules using the collision limit and thermodynamic stability of iodide ion adducts, Atmos Meas Tech, 9(4), 1505–1512, doi:10.5194/amt-9-1505-2016, 2016.

---

## Referee Comment (RC2) · Anonymous Referee #3 · 19 Feb 2018

This paper reported laboratory evaluation of the sensitivities in measuring biogenic volatile organic compounds (BVOC) for chemical ionization mass spectrometry (CIMS) using benzene-derived cations, and discussed the influence of reagent gas concentration, electronic field setting, and water vapor concentration on the instrument's sensitivity. The wide range of tested BVOC, including isoprene, monoterpenes and sesquiterpenes, gave a more comprehensive assessment of the capabilities of this CIMS method to detect BVOC. The ionization and fragmentation pattern of the BVOC were described, including the presence of the [M-1]+ ionization product. The authors showed that after accounting for fragmentation ions, the benzene cluster cation CIMS had comparable sensitivities toward isomers of monoterpenes and sesquiterpenes

across typical boreal forest summer humidity, making the instrument suitable for field quantification of BVOC. This paper contributes to application of CIMS method to BVOC measurement and CIMS ionization methodology, and is recommended for publication in Atmospheric Measurement Techniques after the following comments are addressed.

Page 6 line 161-163: Chloroform is unlikely to be ionized by benzene dimer cation given its higher ionization energy. Where would the chloroform fragment come from, i.e. how is the parent chloroform ion generated?

Page 9 line 237-241: For the unclear benzene cation reaction mechanism with isoprene, can you approach it through relationship between sharing pi electrons and reaction enthalpy? Presumably, the isoprene molecule shares its pi electrons with the benzene cation. For bigger benzene cation clusters, they have bigger pi system and more pi electrons, and isoprene needs to share "less" of its pi electrons with the benzene cation. It seems reasonable this trend with increasing benzene concentrations was observed.

Page 10 line 268-271: The cited work by Ibrahim et al. (2005) does not contain any IR spectrum. It is also hard to imagine a 3-body deprotonation process that involves benzene, water cluster and isoprene, as proposed by the paper. Could m/z 73 be an isobaric ion of water tetramer ion? Does the ToF have the resolution to determine the exact mass and identify the chemical formula? Also, given this high intensity of protonated water clusters ($\sim$9E4 Hz water ion in Figure 6, comparable to 2E5 Hz benzene ion in Figure 5), could BVOC also undergo proton transfer reaction in the IMR?

Page 10 line 280: It is not self-explanatory how instrument operational configuration (benzene concentration and electric field) would cause the inconsistency between current work and Kim et al. (2016). More clarifications are needed here.

Page 10 line 292-294: If limonene is ionized through charge transfer followed by isomerization, how to rationalize the fact the stronger C-H bond is broken, instead of the weaker C-C bond? What hydride abstraction reactions for alkenes have been reported

in literature that can be related to this work?

Figure 3 caption line 370-371: "...using a liquid reagent ion delivery..." should be "liquid reagent ion precursor delivery". "...the first RF-only octupole..." I assume it is RF-only quadrupole here. Also in both panels in figure 3, the peak at m/z 156 should be $(C_6H_6)+(C_6H_6)$, not the trimer.

Table 2 and Table 3: The manuscript does not have clear reference to what $f(H_2O)$ and $f(C_6H_6)$ are.

Table 3: The first two ratios under SH=6.9 look like typos.
* * *

---

## Author Comment (AC1) · 9 Mar 2018

**Reviewer 2**

1) The manuscript states that a "new set of applied voltages with softer de-clustering power" (line 118) is used here, however there is no information given on what the voltages are or how the voltages compare to those used by the group previously (other than the above vague remark). For instance, there is no discussion about how the relative intensity of the $C_6H_6^+$ and $(C_6H_6)^+(C_6H_6)$ compare. Given the current emphasis on voltage scanning within the community (Brophy and Farmer, 2016; Lopez-Hilfiker et al., 2016), I am surprised that such experiments were not carried out here. I think that the inclusion of such experiments would be an asset to the manuscript as it may help provide (at least a basic) explanation of some of the peculiarities that are occurring with isoprene. I am not suggesting that the declustering experiments be performed for all the compounds investigated here; however, the inclusion of a select few would improve the impact of the manuscript. Currently, the only change in the declustering voltages is by turning off the voltages in the first RF-only quadrupole and these results are only shown for sampling zero air (no analyte; Figure 3). While this provides some insight, it is limited given the large drop in ion transmission and the lack of measurement of an analyte. It would be interesting to see how the distribution of $[M-1]^+$, $M^+$, and $[M+32]^+$ changed as a function of voltage scanning for the monoterpenes and sesquiterpenes. I think it is also an overstatement to say in the abstract (lines 19 & 22) and conclusion (line 328) that the effect of declustering voltages was studied given the limited investigation into this effect.

The reviewer raises an important point that highlights we need to further clarify our intent regarding voltage settings. In Bertram et al. [2011], we described the utility of voltage tuning of the exit lenses of the RF-only octupoles to vary cluster distributions while maintaining high transmission. The operational voltages used here have essentially the same field strengths at the "weak field" described in Bertram et al. [2011]. The only deviation is that we start from 0V on the front end of the IMR for practical considerations.

To reduce field strengths even further, in an attempt to probe the cluster distributions of the IMR, we turned off the RF-only quadrupoles completely. This is not meant as an operational configuration, as ion transmission is sacrificed (as the reviewer notes). However, it gives us the best insight into the cluster distribution in the IMR and hence the ion-molecule reaction mechanism. Below, we show the impact of the RF-only quadrupole on analyte-cluster distributions for DMS and isoprene.

[Figure]

As shown above, reduction in the field strength favors the M+78 cluster in the case of DMS, as expected. In the case of isoprene, with the focusing RF quad on, we observe a small percentage of isoprene as the

bare ion (ca. 10%). We expect that this is due to declustering of the isoprene-benzene cluster at a point when a sufficient amount of positive charge is on the isoprene in the cluster. Following declustering, there are few collisions of the bare isoprene ion with benzene and the molecular ion is transferred to the mass analyzer. With the RF-only quad off, we do not observe statistically significant amounts of the bare ion. This suggests that the isoprene is found exclusively as the cluster in the IMR.

The manuscript has been edited in the following locations to address these points:

Line 18: In addition, we determine the dependence of CI-ToFMS sensitivity on the reagent ion neutral delivery concentration and water vapor concentration.

We have also removed the phrase "declustering voltages" from line 328 as the reviewer is correct that we adjusted the field strength to learn about the ionization mechanism, but not to develop new operational configurations for sampling.

Line 118: This line has been edited to now read: "We also examine the de-clustering power of the RF only quadrupole to better determine the cluster distribution present in the ion molecule reaction chamber."

2) Sections 3.1 & 3.2: How does the abundance and ratio between $C_6H_6^+$ and $(C_6H_6)^+(C_6H_6)$ change as the concentration of neutral benzene is increased? Does this provide any insight into the isoprene results?

We expect that at higher benzene neutral concentrations, larger benzene cluster cations $((C_6H_6)^+(C_6H_6)_n$, $n>1)$ are favored. However, we do not have evidence for this as clusters larger than n=1 have very weak binding enthalpies, prohibiting their efficient transmission in the mass spectrometer. As noted in the manuscript, even with the RF-only quadrupole off, we do not observe clusters larger than n=1. As a result, there is only a small change in the cluster distribution observed in the mass spectra. The effect of benzene neutral concentration on the observed cluster distribution and the total benzene cluster ion concentration are shown below. We note that at low benzene neutral concentrations, a significant fraction of total ion current can be found as protonated water clusters or $O_2^+$. We caution over interpretation of the reduction in total benzene cluster ion signal intensity at low benzene neutral concentrations (< 50 ppm) shown here as this experiment was not designed to assess the low concentration regime.

[Figure]

3) I do not see where the text references the tables. Including a reference to the table in the text would be beneficial since the tables show that the M+, [M-1]$^+$, and [M+32]$^+$ ions were investigated for all the compounds and not just those discussed in depth in the text.

We have added references to the tables in the text.

The manuscript has been edited in the following locations to address these points:

Table 1 is now referenced in line 113: "We discuss the sensitivity of benzene cluster cation chemistry to a select number of terpenes (Table 1) at atmospherically relevant mixing ratios (<500 pptv)."

Table 1 is now referenced in line 304: "The reported sensitivities, product ions, and dependence on ambient water concentrations and neutral benzene concentration for select monoterpenes are shown in Table 2."

Table 1 is now referenced in line 323: "The reported sensitivities, product ions, and dependence on ambient water concentrations and neutral benzene concentration for select sesquiterpenes are shown in Table 3."

4) Line 14: replace alpha with the correct symbol

This has been corrected.

5) Line 57-58: The references for $NO^+$ and $H_3O^+$ seem a bit strange as the references refer only to early work and not more recent work, while the references for other ionization mechanisms contains more recent work. For instance, the absence of Koss et al. (2016).

References to Karl et al., 2012, Mochalski et al., 2014, and Koss et al., 2016 have been added.

6) Line 185: "error bars are the standard deviation of repeated triplicate measurements" but caption of figure 2 "error bars represent the standard deviation of the 1 Hz measurements."

The figure caption has been revised. The error bars represent the standard deviation of the triplicate calibrations.

7) Line 209: "Without the first RF-only quadrupole" Is it removed or just the voltages turned off. If the voltages was it all the voltages or just the RF component?

The manuscript has been revised to read: " … but without the RF and voltage bias applied to the first quadrupole ion guide."

8) Line 330-331: The wording here makes it appear like a recommendation to only use $C_{10}H_{16}^+$ for quantification of monoterpenes. However, the average sensitivity appears to be that obtained using the $M^+$, $[M-1]^+$, and $[M+32]^+$ ions (Figure 10) and not just the $M^+$ ion (Figure 7). What recommendation do the authors provide for quantification of monoterpenes?

As shown in Table 2 (and 3), for most monoterpenes and sesquiterpenes measured, the majority of the measured signal is as the molecular ion. However, select monoterpenese (and sesquiterpenes) such as D-limonene have a significant amount of signal at [M-1] and [M+32]. For these reasons, we would recommend including signal intensity at these masses in the overall measurement of monoterpenes (and sesquiterpenes).

The manuscript has been revised (line 337) to read: "Given that signal intensity was observed at ($[M-1]^+$ and ($[M+32]^+$, for a few select terpenes (e.g., D-limonene) we recommend that future measurements of total monoterpenes utilize all three product ions."

9) Lines 333-334: Same comment as above expect for sesquiterpenes.

See response to comment #8.

10) Table 1: I don't find this particularly useful since there is no in depth exploration of how structure might relate to the different observed sensitivities. To be clear, I am not suggesting that such an explanation be incorporated; I do not think it would be appropriate given the extent of the data.

While we did not make tangible connections between terpene structure and sensitivity, we think the table is helpful to the reader that may not be directly familiar with terpene structure.

11) Figure 3: The peak at m/z 156 should be labeled $(C_6H_6)^+$ $(C_6H_6)$

Thank you for catching this error. The figure has been revised.

---

## Author Comment (AC2) · 9 Mar 2018

**Reviewer 3**

1) Page 6 line 161-163: Chloroform is unlikely to be ionized by benzene dimer cation given its higher ionization energy. Where would the chloroform fragment come from, i.e. how is the parent chloroform ion generated?

We expect that the chloroform fragment is generated from trace $O_2^+$ in the instrument. The sensitivity is extremely small (ca. 100 Hz for 180 ppm).

2) Page 9 line 237-241: For the unclear benzene cation reaction mechanism with isoprene, can you approach it through relationship between sharing pi electrons and reaction enthalpy? Presumably, the isoprene molecule shares its pi electrons with the benzene cation. For bigger benzene cation clusters, they have bigger pi system and more pi electrons, and isoprene needs to share "less" of its pi electrons with the benzene cation. It seems reasonable this trend with increasing benzene concentrations was observed.

This is possible. We decided not to speculate on the dependence of the isoprene sensitivity on the concentration of neutral benzene in the manuscript. We expect, although do not have evidence, that the benzene clusters are larger for higher neutral benzene concentrations. It is possible that for larger benzene clusters a larger fraction of the charge is on isoprene at the point of collisional declustering as the reviewer suggests. This would translate to a larger sensitivity at higher benzene concentration. We are working on a subsequent paper that focuses on quantum calculations of these benzene adducts and elect not to speculate on the cause in this manuscript.

3) Page 10 line 268-271: The cited work by Ibrahim et al. (2005) does not contain any IR spectrum. It is also hard to imagine a 3-body deprotonation process that involves benzene, water cluster and isoprene, as proposed by the paper. Could m/z 73 be an isobaric ion of water tetramer ion? Does the ToF have the resolution to determine the exact mass and identify the chemical formula? Also, given this high intensity of protonated water clusters (~9E4 Hz water ion in Figure 6, comparable to 2E5 Hz benzene ion in Figure 5), could BVOC also undergo proton transfer reaction in the IMR?

We thank the reviewer for catching this citation error. As the reviewer notes, the Ibrahim paper did not contain the IR spectra; it only presented the hydration path of ionized benzene via the arrival time distributions of $C_6H_6\text{-}(H_2O)_n^+$ clusters and supporting quantum calculations. The publication the authors should have also cited is Miyazaki et al. (2004). Miyazaki et al. presents IR spectra of $C_6H_6\text{-}(H_2O)_n^+$ clusters up to n=23. Figures 1 & 3 show that the IR spectra of $C_6H_6\text{-}(H_2O)_n^+$ clusters resemble $H^+(H_2O)_n$ starting at n≥4. Figure 2 of Miyazaki et al. shows the hydrogen bonded $H_2O$ networks required to deprotonate a $C_6H_6^+$. We are speculating that we see the effects of a proton-transfer from a $C_6H_6\cdot C_5H_8^+$ cluster to a network of $H_2O$. At this time we cannot definitively comment on what the structure of that network is exactly but the end product is that of a protonated water tetramer. Our own computational studies have so far shown that a key component of sensitivity in benzene CIMS is through charge migration within a given benzene adduct. The influence of a water environment on that charge migration varies depending on the species in that adduct. We have performed quantum mechanical calculations of charged benzene adducts in the presence of varying $(H_2O)_n$ with water network structures of both those in Ibrahim et al. and Miyazaki et al. and have found that in the case of a $C_6H_6\text{-}C_5H_8^+$ adduct, the positive charge starts to migrate towards a $(H_2O)_5$ cluster. Our group is working on more calculations to prepare a separate paper explaining charge distribution with respect to detection *via* benzene CIMS so we chose not to comment further on this at this time.

We cannot rule out an isobaric ion at m/z 73 with the mass resolution of our ToFMS. However, the sensitivity to the isobaric ion would need to be dependent on water and stem from the isoprene source (to

be consistent with Fig. 5 and 6). It is possible that this is due to a proton transfer as total water cluster intensity will increase with SH, but total water cluster signal even at 14 g kg$^{-1}$ SH is only 3.7E4 Hz, compared with 9.8E5 Hz for benzene clusters. In addition, this would not explain the observed anti-correlation in signal intensity at 73 and 146 m/Q which would be explained by our proposed mechanism.

The reference on line 267 has been changed from Ibrahim et al. (2005) to Miyazaki et al. (2004). We think this sufficiently addresses the reviewer's concerns.

**Reference:** Miyazaki et al., Infrared Spectroscopy of Size-Selected Benzene-Water Cluster Cations [C6H6-(H2O)*n*]+(*n* ) 1-23): Hydrogen Bond Network Evolution and Microscopic Hydrophobicity, *J. Phys. Chem. A* 2004, *108,* 10656-10660

4) Page 10 line 280: It is not self-explanatory how instrument operational configuration (benzene concentration and electric field) would cause the inconsistency between current work and Kim et al. (2016). More clarifications are needed here.

The inconsistency between Kim et al and this paper is in the SH dependence of the α-pinene sensitivity. In Kim et al., we observed an increase in sensitivity with SH. Here, we observe a very slight decrease. We think the difference in benzene neutral concentration is key here. The increase in sensitivity with SH (Kim et al) was observed using three different mass spectrometers for both laboratory calibrations and in field standard additions. All operating with 10 ppm neutral C$_6$H$_6$ concentrations. The slight decrease in sensitivity with SH was replicated over a comprehensive 6 month evaluation of monoterpene sensitivity (this study), at high benzene concentration (>300 ppm).

In Kim et al., we operated at low C$_6$H$_6$ concentrations due to contaminants in the benzene compressed gas source. Unfortunately, this led to an increase in the percentage of protonated water clusters as SH increases. This has a few implications. The most notable is that proper normalization to the primary ion becomes challenging at high SH as the observed signal of benzene cluster cations is small. In Kim et al., we normalized only to the sum of the benzene cluster cations, which may have overestimated the dependence of sensitivity on SH when compared to the current study where the sum of the benzene cluster cation signal is small. We note that this does not impact the conclusions drawn from the analyses of Kim et al. 2016 or 2017, as the normalization method was consistent. However, we strongly recommend that future studies use higher benzene neutral concentrations to avoid these issues.

The manuscript has been revised to state:

Line 279: "This is attributed to the different instrument operational configuration used here (e.g., high concentration and purity benzene reagent ion precursor). More specifically, high neutral benzene concentrations (300 ppmv) suppress the formation of protonated water clusters at high specific humidity. As a result, the benzene cluster cations account for greater than 90% of total ion current under all conditions. This reduces complications surrounding the normalization of analyte signals."

5) Page 10 line 292-294: If limonene is ionized through charge transfer followed by isomerization, how to rationalize the fact the stronger C-H bond is broken, instead of the weaker C-C bond? What hydride abstraction reactions for alkenes have been reported in literature that can be related to this work?

A suggested path would require oxidation of an [M+1]$^+$ ion to form a hydroperoxide followed by HOOH elimination. The presence of water clusters in our system should suggest some proton transfer reactions to limonene to make an [M+1]$^+$. It is known that the autoxidation of *d*-limonene results in the allergens *cis*- and *trans*-limonene-2-hydroperoxides. A hydrogen abstraction can occur following the departure of the

hydroperoxyl radical from one of these peroxides. Given the abundance of limonene hydroperoxides from autoxidation in nature, the absence of [M+1]$^+$, and our sensitivity to the [M+32]$^+$ion, an oxidation product ion, this would make the most sense. This value may decrease with increasing specific humidity due to the change in water cluster size distribution to larger water networks, inhibiting proton donation.

The manuscript has been revised to state:

Line 291: "The peak at 135 $m/Q$ ($C_{10}H_{15}$)$^+$ represents the [M-1]$^+$ product, perhaps due to oxidation of an [M+1]$^+$ ion followed by departure of HOOH (Karlberg et al., 1994)."

Reference: Karlberg et al., *Hydroperoxides in oxidized d-limonene identified as potent contact allergens*, Arch Dermatol Res (1994) 286:97 103

6) Figure 3 caption line 370-371: "...using a liquid reagent ion delivery..." should be "liquid reagent ion precursor delivery". "...the first RF-only octupole..." I assume it is RF-only quadrupole here. Also in both panels in figure 3, the peak at m/z 156 should be (C6H6)+(C6H6), not the trimer.

This has been changed.

7) Table 2 and Table 3: The manuscript does not have clear reference to what f(H2O) and f(C6H6) are.

Line 304 and 323 have been edited to include these definitions: "The reported sensitivities, product ions, and dependence on ambient water concentrations, f($H_2O$), and neutral benzene concentration, f($C_6H_6$), for select monoterpenes are shown in Table 2."

8) Table 3: The first two ratios under SH=6.9 look like typos.

Thank you, this has been updated.

---

## Author Response (AR2)

**Response to Reviewers Comments:**

**Below, please find our responses to the reviewers second round of comments:**

One question in reading the revised manuscript: lines 298-299 mentions that the [M+32]+ peak is reduced by sampling in N2. Is the same observed for the [M-H]+ peak that the authors speculate could be formed via oxidation of the [M+1]+ ion? If not, can the authors provide a different explanation for the [M-H]+ peak?

Yes, the  $[M-1]^+$  peak intensity is also reduced to near zero when sampling N2. This has been noted on line 298 in the revised manuscript.

Figure 9: The symbols in the figure do not match the description in the text.

This has been edited.

Figure 5 on page 26 should be figure 12. The caption doesn't match the symbols used.

This has been edited.

Figure 6 on page 27 should be figure 13

This has been edited.

[revised manuscript text omitted]